# Support received after bereavement by suicide and other sudden deaths: a cross-sectional UK study of 3432 young bereaved adults

Alexandra L Pitman,[1,2] Khadija Rantell,[3] Paul Moran,[4] Lester Sireling,[5] Louise Marston,[6] Michael King,[1,2] David Osborn[1,2]

► Prepublication history and additional material are available. To view these files please visit the journal online (http://dx.doi.org/10.1136/bmjopen-2016-014487).

[1]UCL Division of Psychiatry, University College Medical School, London, UK
[2]Camden and Islington NHS Foundation Trust, St Pancras Hospital, London, UK
[3]Institute of Neurology, University College London, London, UK
[4]University of Bristol, Bristol, UK
[5]Independent medico-legal practice, London, UK
[6]Primary Care and Population Health, University College Medical School, London, UK

**Correspondence to**
Dr Alexandra L Pitman;
a.pitman@ucl.ac.uk

## ABSTRACT

**Objective** To test the hypothesis that people bereaved by suicide are less likely to receive formal or informal support than people bereaved by other causes of sudden death.

**Design** National cross-sectional study.

**Setting** Adults working or studying at any UK higher education institution (HEI) in 2010.

**Participants** A total of 3432 eligible respondents aged 18–40 years bereaved by the sudden death of a close friend or relative, sampled from approximately 659 572 bereaved and non-bereaved staff and students at 37 of 164 UK HEIs invited to participate.

**Exposures** Bereavement by suicide (n=614; 18%), by sudden unnatural causes (n=712; 21%) and by sudden natural causes (n=2106; 61%).

**Main outcome measures** Receipt of formal and informal support postbereavement; timing of valued support.

**Results** 21% (725/3432) of our sample of bereaved adults reported receiving no formal or informal bereavement support, with no evidence for group differences. People bereaved by suicide were less likely to have received informal support than those bereaved by sudden natural causes (adjusted OR (AOR)=0.79; 95% CI 0.64 to 0.98) or unnatural causes (AOR=0.74; 95% CI 0.58 to 0.96) but did not differ from either comparison group on receipt of formal support. People bereaved by suicide were less likely to have received immediate support (AOR=0.73; 95% CI 0.59 to 0.90) and more likely to report delayed receipt of support (AOR=1.33; 95% CI 1.08 to 1.64) than people bereaved by sudden natural causes. Associations were not modified by gender, or age bereaved, but became non-significant when adjusting for stigma.

**Conclusions** People bereaved by suicide are less likely to receive informal support than people bereaved by other causes of sudden death and are more likely to perceive delays in accessing any support. This is concerning given their higher risk of suicide attempt and the recommendations within suicide prevention strategies regarding their need for support.

**Study registration** http://www.ucl.ac.uk/psychiatry/bereavementstudy/

## INTRODUCTION

Empirical research now supports an association between bereavement by suicide and a

---

### Strengths and limitations of this study

► We conducted a large population-based closed survey to identify bereaved friends and relatives, avoiding the biases inherent to using a help-seeking sample.
► We captured use of a wide range of formal and informal support sources and the time taken to access valued support.
► We compared support use after different modes of sudden bereavement to test a specific hypothesis about inequities in support for people bereaved by suicide.
► Given the age range sampled and the possibility of selection bias (favouring higher social classes) and male non-response bias, the results of this study may only be generalisable to young bereaved women and the more highly educated.

range of negative health outcomes, including an increased probability of suicide[1] and of suicide attempt in close contacts.[2] US and UK suicide prevention strategies recommend providing support for people bereaved by suicide,[3–6] but the extent of implementation is unknown. The starting point in addressing this task is to provide a description of the nature of support services currently used. The next challenges are the paucity of trial evidence for effective interventions[7] and the tendency of people bereaved by suicide to avoid seeking help[8 9] despite expressing clear unmet needs.[10] This avoidance is likely to be linked to stigmatising societal beliefs about suicide as a failure of problem solving.[9] High levels of stigma relative to other bereaved groups[11] may reduce both willingness to seek help and friends' or relatives' readiness to offer support.[9 12] This is concerning if stigma adversely affects access to support in a population vulnerable to suicide.[13]

No British study has provided an overview of the range of support received by people

bereaved by suicide. US surveys have tended to be small and localised[14–16] or involve help-seeking samples.[17] Registry-based studies describe health service use[18–21] but not informal support, a resource known to be valued after suicide bereavement.[10] For service planning purposes, we lack population-based studies describing the prevalence and correlates of support received by people bereaved by suicide. Our objective was to address this by conducting a nationwide population-based survey of bereaved adults, collecting data on health outcomes and support received after sudden bereavement. We focused on young adults given concerns about their vulnerabilities to suicide,[22] their tendency to avoid accessing mental health services[23] and their priority status within UK suicide prevention strategies.[4–6] Surveying this age range also minimised the potential for memory decay and narrowed period effects. We aimed to answer the following research questions about people bereaved by suicide, compared with those bereaved by other sudden forms of death: whether they are less likely to receive formal and informal support and more likely to receive no support or delayed support; whether they are more likely to rely exclusively on formal support; whether perceived stigma accounts for reduced receipt of support and whether there are gender differences in support received.

## METHOD

### Patient involvement

Our research question was prompted by UK suicide prevention strategies[4–6] and developed in consultation with a group of bereaved adults and bereavement counsellors. This consultation group identified important outcomes to capture in relation to the impact of sudden bereavement and provision of support and reviewed successive drafts of the survey questionnaire. This questionnaire was piloted with individuals accessing support from four national bereavement support organisations: Cruse Bereavement Care, Samaritans, Survivors of Bereavement by Suicide and Widowed by Suicide. Patients were not involved in the population-based recruitment of this study or data analysis. All bereaved individuals participating in the survey were invited to provide contact details for dissemination of study findings and to bookmark the findings section of the study website: http://www.ucl.ac.uk/psychiatry/bereavementstudy.

### Study design and participants

We conducted a national cross-sectional survey of young adults working or studying at UK higher education institutions (HEIs), avoiding the biases associated with recruiting a help-seeking sample.[24] In 2010, all 164 HEIs in the UK were invited to participate, following up non-responding HEIs to encourage broad socioeconomic and geographic representation. Over 20% (37/164) agreed, with a higher response (40%) from those classified as the more prestigious Russell Group universities. This accessed an estimated sampling frame of 659 572

staff and students. The majority of participating HEIs followed study protocol in sending an individual email invitation, with embedded survey link, to each staff and student member. For reasons of sensitivity, 10 HEIs modified this strategy, either by emailing students only, using their weekly news digest email or advertising via staff and student intranet. All recipients, whether bereaved or not, were invited to take part in a survey of 'the impact of sudden bereavement on young adults', with the aim of masking them to the specific study hypotheses. As the denominator of bereaved people could not be ascertained using survey methods or routine data, there was no accurate way of measuring the proportion of bereaved people who responded.

Inclusion criteria were people aged 18–40 years who had experienced sudden bereavement of a close friend or relative. Early childhood bereavements (before age 10) were excluded to minimise recall bias. Sudden bereavement was defined as 'a death that could not have been predicted at that time and which occurred suddenly or within a matter of days'. Exposure status was subclassified by self-report as: bereavement by suicide, bereavement by sudden unnatural causes (eg, accidental death) and bereavement by sudden natural causes (eg, cardiac arrest). For respondents who had experienced more than one type of sudden bereavement, we categorised exposure as follows: all those bereaved by suicide were classified as such, regardless of other exposures. Those bereaved by non-suicide death were asked to relate their responses to whichever person they had felt closest to, with exposure status classified accordingly. We based our sample size calculation on the primary outcome for a separate study investigating the association between suicide bereavement and suicide attempt,[2] indicating that at least 466 participants were required in any one group (two-tailed analysis; 90% power).

The study was approved by the University College London (UCL) Research Ethics Committee in 2010 (ref: 1975/002). All participants provided online informed consent.

### Procedures

Our online questionnaire[2] elicited quantitative data on sociodemographic and clinical characteristics. We described past suicidal ideation, suicide attempt and non-suicidal self-harm using standardised measures from the Adult Psychiatric Morbidity Survey,[25] which distinguishes suicide attempt from non-suicidal self-harm on the basis of intent.[26] We qualified whether each had occurred before or after the bereavement or both. Depression was measured using the Composite International Diagnostic Interview screen for lifetime depression,[27] qualified as above. Perceived stigma, the subjective awareness of others' stigmatising attitudes, was measured using the stigma subscale of the Grief Experience Questionnaire.[28] Likert-style responses to 10 items (eg, 'Since the death how often did you feel avoided by friends?') generated scores of 5–25. We used a fixed-choice question to

ascertain the stage at which respondents felt they had been most affected by the loss.

Two tick-box questions probed help received, whether sought or offered, after the bereavement: 'How long after the death did you receive help that was valuable to you?' and 'What help did you receive after the death?' (with 10 options, including None and Other—please state). Two tick-box questions probed help-seeking for self-harm: 'If you have harmed yourself since the bereavement did you seek help from anyone?' and 'Who did you try to get help from?' (with five options, including Other—please state). We derived our seven binary outcomes from responses to these questions.

Our two primary outcomes were: receipt of any formal bereavement support and receipt of any informal bereavement support. Formal and informal support classifications were derived from similar British[23] and international studies of service use.[29] Self-help was considered a separate category due to problematic formal/informal categorisation in relation to bereavement support.[30]

Four secondary outcomes were: receipt of no valuable support, immediate receipt (within 1 week) of valuable support, delayed receipt (beyond 6 months) of valuable support and exclusive use of formal support. These thresholds were agreed on the basis of clinical judgement and the published literature.[31] A fifth secondary outcome was whether those who had attempted suicide postbereavement had sought help for this.

### Statistical analysis

We summarised sample characteristics by exposure group using $\chi^2$ testing (categorical variables) and one-way analysis of variance (continuous variables). We used multivariable random effects logistic regression to estimate the strength of the associations between mode of bereavement exposure (sudden natural causes/sudden unnatural causes/suicide) and binary outcomes. Our multivariable models included eight prespecified confounding variables identified from existing literature and clinical judgement: age, gender, socioeconomic status, preloss depression, preloss suicidal and non-suicidal self-harm, other family history of suicide (excluding index suicide bereavement), time since bereavement and kinship to the deceased. We used HEI as random effect to take account of clustering effects at the institutional level.

For each outcome we conducted two distinct comparisons. The first controlled for the sudden nature of the death, using people bereaved by sudden natural causes as reference category. The second controlled for the violence of the death, using people bereaved by sudden unnatural causes as reference category.

To test whether stigma attenuated any associations between bereavement exposure and outcomes, we added perceived stigma[28] to our final models.

We added an interaction term to our final models to test a further prespecified hypothesis: that the effect of bereavement on receipt of support varied by gender such that men bereaved by suicide would show a more marked lack of formal and informal support. In a post hoc test for interaction, we assessed whether age at bereavement (before or after age 18) influenced receipt of support, such that bereaved children would be better supported.

Finally, we conducted a priori sensitivity analyses to assess the impact of simulating predicted non-response biases, excluding 918 respondents from the 10 HEIs that had modified the protocol recruitment method. We conducted an additional sensitivity analysis to compare people bereaved by suicide to a reference category of all those bereaved by non-suicide sudden death.

All analyses were conducted using Stata V.12[32] and complete case analysis.

### RESULTS

Of the 659 572 bereaved and non-bereaved people invited to take part, 5085 people responded to the questionnaire by clicking on the survey link. Of these, 91% (n=4630) consented to participate and 68% (n=3432) fulfilled eligibility criteria (figure 1). Cluster (HEI) size varied from 3 to 364 participants (median=52; IQR=25–120). Missing data for model covariates and outcomes were less than 7% for covariates and less than 4% for outcomes.

The sample was primarily female, white and related to the deceased by blood (table 1). There were no statistically significant group differences by bereavement exposure in relation to gender, age, ethnicity, socioeconomic status, personality disorder screen[33] or perceived level of social support. The mean time elapsed since bereavement was 4.9 years (SD=5.3; range=1 day–30 years), with no significant group differences. One quarter (24%; 824/3432) reported that they had been most affected in the first week after the loss, but a third (38%; 1274/3432) endorsed over 6 months after the loss, with no evidence for group differences.

Overall, 78% (2572/3432) of the sample reported receiving some form of support after the loss, whether informal (51%), formal (14%) or both (35%), and 85% (2173/2572) perceived some aspect of it to have been valuable. Two-fifths (42%; 1438/3432) had received valuable support within a week of the loss. Overall, 20% of the sample received no support at all, excluding the 20 individuals who specified that they had chosen to handle the bereavement alone. The most endorsed source of informal support was family and friends (64%) and of formal support were funeral directors (14%) and health professionals (13%). Self-help was used by 10% (table 2).

Overall 6% reported having attempted suicide since the bereavement, of which 67% (141/210) had not sought help for any episode of self-harm occurring postbereavement (table 2). In those who had sought help, the most common source was a general practitioner (20%).

People bereaved by suicide were significantly less likely to receive informal support than those bereaved by sudden natural causes (table 3; adjusted OR (AOR)=0.79;

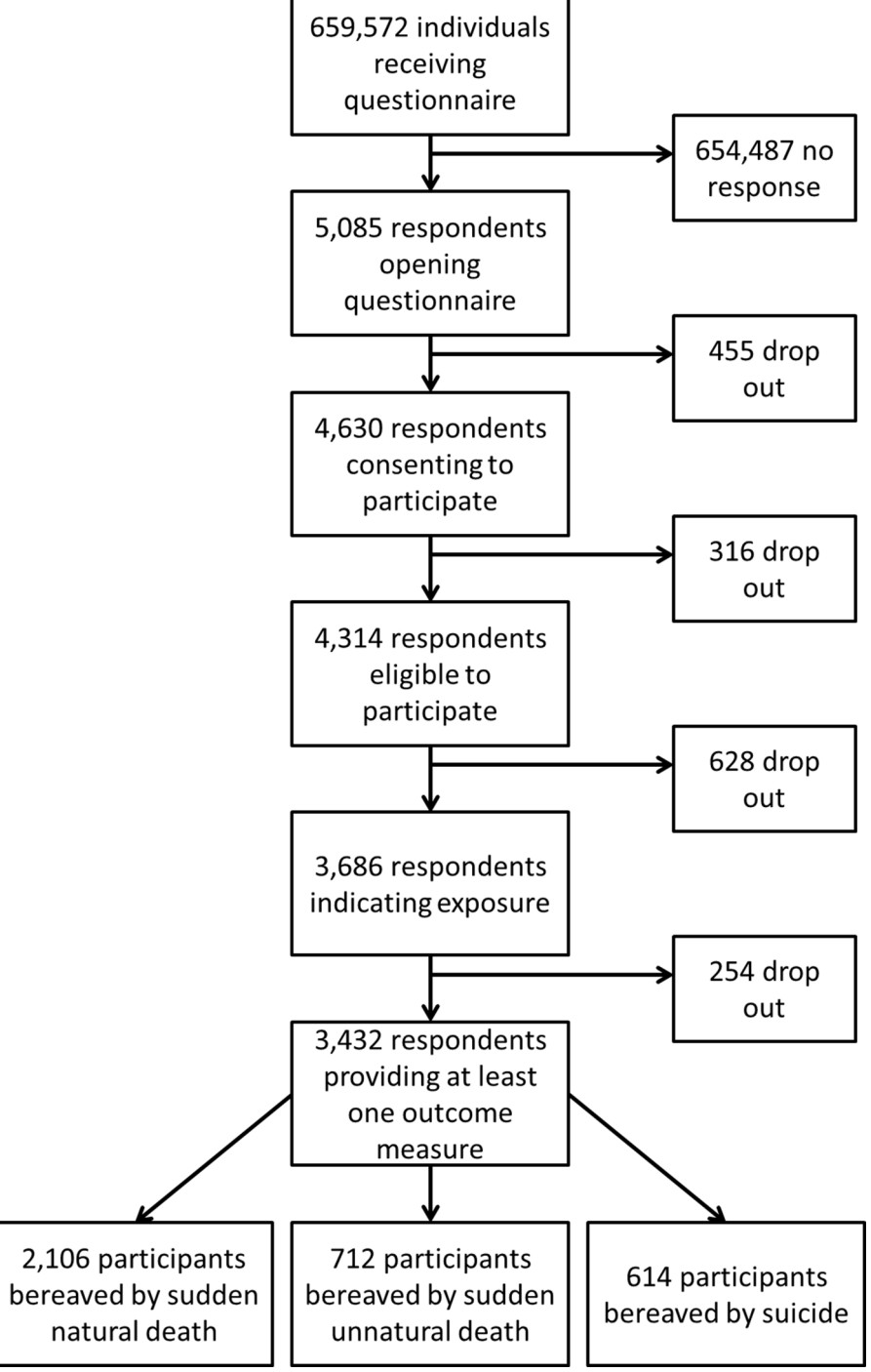

**Figure 1** Participant flow.

95% CI 0.64 to 0.98) and those bereaved by unnatural causes (table 4; AOR=0.74; 95% CI 0.58 to 0.96). People bereaved by sudden unnatural causes were significantly more likely to receive formal bereavement support than those bereaved by sudden natural causes (table 3; AOR=1.28; 95% CI 1.05 to 1.56), but there were no other group differences on this outcome.

Compared with people bereaved by sudden natural mortality causes, people bereaved by suicide were significantly less likely to receive immediate support (table 3; AOR=0.73; 95% CI 0.59 to 0.90) and significantly more

likely to report delayed receipt of support (AOR=1.33; 95% CI 1.08 to 1.64). There were no other group differences on this or any other secondary outcome. After adding perceived stigma to models, all four significant associations of suicide bereavement with support outcomes became non-significant, as did the association between bereavement by sudden unnatural causes and use of formal bereavement support.

Interaction tests showed no evidence that gender, or childhood versus adult bereavement, modified any of the associations identified.

**Table 1** Characteristics of participants by type of bereavement exposure

| Participants bereaved by: | Sudden natural death (n=2106) | Sudden unnatural death (n=712) | Suicide (n=614) | Total (n=3432) | p Value* |
|---|---|---|---|---|---|
| **Sociodemographic characteristics** | | | | | |
| Gender (n (%))† | | | | | |
| Female | 1709 (81) | 576 (81) | 499 (81) | 2784 (81) | 0.955 |
| Missing | 1 (<1) | 0 (0) | 0 (0) | 1 (<1) | |
| Age of participant† mean (SD) | 24.9 (6.3) | 25.2 (6.3) | 25.2 (6.0) | 25.0 (6.3) | 0.069 |
| Self-defined ethnicity (n (%)) | | | | | |
| White | 1877 (89) | 645 (91) | 562 (92) | 3084 (90) | 0.102 |
| Missing | 1 (<1) | 2 (<1) | 0 (0) | 3 (<1) | |
| **Socioeconomic status (n (%))†‡** | | | | | |
| Social classes 1.1 and 1.2 | 603 (29) | 224 (32) | 176 (29) | 1003 (29) | 0.604 |
| Social class 2 | 684 (33) | 234 (33) | 204 (33) | 1122 (33) | |
| Social class 3 | 259 (12) | 77 (11) | 68 (11) | 404 (12) | |
| Social class 4 | 90 (4) | 34 (5) | 32 (5) | 156 (5) | |
| Social classes 5, 6, 7 and 9 | 409 (19) | 115 (16) | 113 (18) | 638 (19) | |
| Missing | 61 (3) | 27 (4) | 21 (3) | 109 (3) | |
| **Educational status (n (%))** | | | | | |
| Attained up to secondary school leaving qualification | 964 (46) | 286 (40) | 255 (42) | 1505 (44) | **0.035** |
| Attained degree or above | 1136 (54) | 424 (60) | 359 (59) | 1919 (56) | |
| Missing | 6 (<1) | 2 (<1) | 0 (0) | 8 (<1) | |
| **Student status (n (%))** | | | | | |
| Student | 1797 (85) | 613 (86) | 526 (86) | 2936 (86) | 0.905 |
| Staff | 253 (12) | 78 (11) | 68 (11) | 399 (12) | |
| Both | 55 (3) | 21 (3) | 20 (3) | 96 (3) | |
| Missing | 1 (<1) | 0 (0) | 0 (0) | 1 (<1) | |
| **Measure of social support (n (%))§** | | | | | |
| No lack of perceived social support | 1234 (59) | 411 (58) | 345 (56) | 1990 (58) | 0.297 |
| Moderate lack of perceived social support | 549 (26) | 197 (28) | 168 (27) | 914 (27) | |
| Severe lack of perceived social support | 323 (15) | 102 (14) | 100 (16) | 525 (15) | |
| Missing | 0 (0) | 2 (<1) | 1 (<1) | 3 (<1) | |
| **Clinical characteristics** | | | | | |
| **Personality disorder screen positive (n (%))¶** | | | | | |
| Yes | 743 (35) | 227 (32) | 225 (37) | 1195 (35) | 0.071 |
| Missing | 131 (6) | 31 (4) | 33 (5) | 195 (6) | |
| Family history of psychiatric problems (n (%)) | | | | | |
| Yes | 1243 (59) | 434 (61) | 412 (67) | 2089 (61) | **0.005** |
| Missing | 153 (7) | 41 (6) | 39 (6) | 233 (7) | |
| Other family history of suicide (n (%))† | | | | | |
| Yes | 123 (6) | 41 (6) | 53 (7) | 217 (6) | 0.071 |
| Missing | 158 (8) | 43 (6) | 40 (7) | 241 (7) | |
| Preloss non-suicidal self-harm and suicide attempt (n (%))† | | | | | |
| Yes | 434 (21) | 134 (19) | 150 (24) | 718 (21) | 0.050 |
| Missing | 157 (8) | 41 (6) | 41 (7) | 239 (7) | |

Continued

**Table 1** Continued

| Participants bereaved by: | Sudden natural death (n=2106) | Sudden unnatural death (n=712) | Suicide (n=614) | Total (n=3432) | p Value* |
|---|---|---|---|---|---|
| **Preloss depression (n (%))†** | | | | | |
| Yes | 370 (18) | 129 (18) | 143 (23) | 642 (19) | **0.015** |
| Missing | 85 (4) | 21 (3) | 24 (4) | 130 (4) | |
| **Characteristics of the bereavement** | | | | | |
| **Kinship to the deceased (n (%))†** | | | | | |
| Blood relative | 1786 (85) | 351 (49) | 296 (48) | 2433 (71) | **<0.001** |
| Unrelated | 313 (15) | 356 (50) | 317 (52) | 980 (29) | |
| Missing | 7 (<1) | 5 (1) | 1 (<1) | 13 (<1) | |
| Age of the deceased (mean (SD)) | 55.1 (21.5) | 31.0 (17.4) | 31.9 (15.2) | 45.9 (22.8) | **<0.001** |
| Time since bereavement† (mean (SD)) | 4.8 (5.3) | 5.3 (5.4) | 5.1 (5.0) | 5.0 (5.3) | 0.140 |
| **Bereavement in last 2 years (n (%))** | | | | | |
| Yes | 707 (34) | 186 (26) | 168 (27) | 1061 (31) | **<0.001** |
| No | 1399 (67) | 526 (74) | 446 (73) | 2371 (69) | |
| GEQ stigma subscale score (mean (SD)) | 11.9 (3.8) | 12.3 (4.0) | 14.0 (4.3) | 12.3 (4.0) | **<0.001** |
| **Time point rated as worst stage after the loss (n (%))** | | | | | |
| Within a week | 560 (25) | 156 (22) | 148 (24) | 824 (24) | 0.112 |
| Up to a month | 330 (16) | 92 (13) | 81 (13) | 503 (15) | |
| Up to 6 months | 330 (16) | 122 (17) | 112 (18) | 564 (16) | |
| Up to a year | 359 (17) | 147 (21) | 101 (17) | 607 (18) | |
| Up to 3 years | 216 (10) | 80 (11) | 69 (11) | 365 (11) | |
| Over 3 years | 181 (9) | 62 (9) | 59 (10) | 302 (9) | |
| Missing | 170 (8) | 53 (8) | 44 (7) | 267 (8) | |
| **Bereavement support** | | | | | |
| **Any formal/informal support†† received after bereavement (n (%))** | | | | | |
| Yes | 1573 (75) | 558 (78) | 441 (72) | 2572 (75) | **0.031** |
| No | 446 (21) | 131 (18) | 148 (24) | 725 (21) | |
| Missing | 87 (4) | 23 (3) | 25 (4) | 135 (4) | |
| **Formal/informal support perceived to be valuable (of n=2572) (n (%))** | | | | | |
| Yes | 1335 (85) | 464 (83) | 374 (85) | 2173 (85) | 0.621 |
| No | 216 (14) | 85 (15) | 59 (13) | 360 (14) | |
| Missing | 22 (1) | 9 (2) | 8 (2) | 39 (2) | |
| **Type of formal/informal support received (of n=2572) (n (%))** | | | | | |
| Formal only | 217 (14) | 76 (14) | 68 (15) | 361 (14) | 0.922 |
| Informal only | 796 (51) | 286 (51) | 220 (50) | 1302 (51) | |
| Both formal and informal | 560 (36) | 196 (35) | 153 (35) | 909 (35) | |
| **Point at which any valuable support received after loss (n (%))** | | | | | |
| Within a day | 623 (30) | 234 (33) | 150 (24) | 1007 (29) | **0.001** |
| Within a week | 290 (14) | 72 (10) | 69 (11) | 431 (13) | |
| Within a month | 154 (7) | 50 (7) | 44 (7) | 248 (7) | |
| Within 6 months | 117 (6) | 35 (5) | 46 (8) | 198 (6) | |
| Within a year | 58 (3) | 31 (4) | 15 (2) | 104 (3) | |

**Table 1** Continued

| Participants bereaved by: | Sudden natural death (n=2106) | Sudden unnatural death (n=712) | Suicide (n=614) | Total (n=3432) | p Value* |
|---|---|---|---|---|---|
| Over a year | 124 (6) | 49 (7) | 58 (10) | 231 (7) | |
| At no point | 632 (30) | 211 (30) | 198 (32) | 1041 (30) | |
| Missing | 108 (5) | 30 (4) | 34 (6) | 172 (5) | |
| Whether help sought after self-harm postbereavement (n (%))‡‡ | | | | | |
| Yes | 42/112 (38) | 8/42 (19) | 19/56 (34) | 69/210 (33) | 0.093 |
| No | 70/112 (63) | 34/42 (81) | 37/56 (66) | 141/210 (67) | |

*Significance threshold of p=0.05; not adjusted for multiple testing.
†Prespecified confounding variable used in adjusted models.
‡Socioeconomic status using the five categories from UK Office for National Statistics.
§Measure of social support from Adult Psychiatric Morbidity Survey.[25]
¶Self-report Standardized Assessment of Personality-abbreviated Scale (SAPAS-SR) screen for personality disorder.[33]
††Excluding self-help.
‡‡In subsample of n=210 who had made a suicide attempt since the index bereavement.
GEQ, Grief Experience Questionnaire.

In sensitivity analyses simulating predicted non-response biases, the magnitude and direction of significant associations between suicide bereavement and outcomes were unchanged, apart from the association between bereavement by sudden unnatural causes and use of formal bereavement support, which became non-significant. In an analysis comparing suicide bereavement to all non-suicide sudden bereavements, we found similar associations in terms of magnitude and direction, apart from the association of suicide bereavement with one secondary outcome (delayed receipt of valuable support), which became non-significant (see online supplementary table).

## DISCUSSION
### Main findings
One in four people bereaved by suicide in this national sample had received no formal or informal support after their loss, despite the major emphasis in English,[4] Northern Irish[5] and Welsh[6] suicide prevention strategies on improved suicide bereavement support. People bereaved by suicide were significantly less likely to have received informal support and more likely to describe delays in receiving any formal or informal support. These findings may not reflect preferences, as receipt of support is a function of what is perceived to be available. It is therefore unclear whether our findings reflect reduced help seeking or an objective lack of help offered. The cross-sectional, observational nature of these data limits causal inference. However, surveys of the perceived needs of people bereaved by suicide indicate clear unmet needs for social networks to respond proactively and empathically and for professionals to offer immediate outreach.[10] This suggests that our findings represent gaps in support rather than a rejection or avoidance of help. Whether stigma explains the inequalities observed, perhaps by inhibiting help-seeking or offers of support, requires further research. The low rates of help seeking after suicide attempt are particularly concerning in people bereaved by suicide given their higher risk of suicide attempt[2] and the high priority accorded to their needs for support within British suicide prevention strategies.

### Results in the context of other studies
Perhaps reflecting cultural differences, our findings differ from those of a representative US sample of suicide-bereaved relatives, in which 24% had received either formal or informal support and 33% preferred to cope without assistance.[34] In a US help-seeking sample, 78% reported receiving individual therapy after suicide bereavement,[17] a proportion greatly exceeding formal support use in our population-based sample. The only British study of support after suicide[35] did not state the overall proportion receiving support, but the prevalence of counselling matched that in our study. Consultation with faith leaders was more common than in our sample (10% versus 2%), perhaps reflecting differing age profiles. Studies comparing groups bereaved by suicide and other causes have only focused on single measures of perceived social support and have, like our study, found weak or no evidence for group differences.[14–16]

### Strengths and limitations
This national sample represents the largest and most comprehensive survey of support received by people bereaved by a close contact's sudden death. It included respondents who were related and unrelated to the deceased, recognising that adverse outcomes and needs for support apply regardless of kinship.[2] In conducting specific group comparisons, we were able to ascertain that reduced receipt of informal support was attributable to suicide bereavement rather than unnatural losses more widely. Results were robust to sensitivity analysis, and use of a precise sampling frame allowed us to be clear about the limits of generalisability. The possibility of selection bias through sampling from

**Table 2** Specific type of support used after bereavement

| Participants bereaved by: | Sudden natural death (n=2106) (n (% of exposure group)) | Sudden unnatural death (n=712) (n (% of exposure group)) | Suicide (n=614) (n (% of exposure group)) | Total (n=3432) (n (% of total sample)) |
|---|---|---|---|---|
| **Specific bereavement support reported*** | | | | |
| Formal support | | | | |
| Health services (doctor, nurse, therapist, counsellor) | 283 (13) | 86 (12) | 83 (14) | 452 (13) |
| Social services | 0 (0) | 0 (0) | 1(<1) | 1 (<1) |
| Private counsellor or therapist | 171 (8) | 78 (11) | 73 (12) | 322 (9) |
| Voluntary sector services (helpline, counsellor) | 120 (6) | 53 (7) | 51 (8) | 224 (7) |
| Police officers | 77 (4) | 102 (14) | 45 (7) | 224 (7) |
| Funeral directors | 359 (17) | 85 (12) | 51 (8) | 495 (14) |
| Coroners' officers | 130 (6) | 51 (7) | 35 (6) | 216 (6) |
| School teachers or school counselling services | 28 (1) | 11 (2) | 9 (2) | 48 (1) |
| College tutor or college counselling services | 34 (2) | 11 (2) | 19 (3) | 64 (2) |
| Line manager or employee counselling services | 5 (<1) | 3 (<1) | 1 (<1) | 9 (<1) |
| *Subtotal formal support* | 1207 (57) | 480 (67) | 368 (60) | 2055 (60) |
| Informal support | | | | |
| Friends and family | 1349 (64) | 481 (68) | 370 (60) | 2200 (64) |
| Spiritual/religious advisors | 40 (2) | 10 (1) | 10 (2) | 60 (2) |
| Complementary and alternative medicine | 1 (<1) | 0 (0) | 0 (0) | 1 (<1) |
| *Subtotal informal support* | 1390 (66) | 491 (69) | 380 (62) | 2261 (66) |
| *Subtotal any formal or informal support* | 1573 (75) | 558 (78) | 441 (72) | 2572 (75) |
| Other | | | | |
| Self-help (website, book, leaflet) | 208 (10) | 61 (9) | 79 (13) | 348 (10) |
| Specific source not specified | 23 (1) | 7 (1) | 6 (1) | 36 (1) |
| Other (not classified as above)† | 3 (<1) | 2 (<1) | 1 (<1) | 6 (<1) |
| *Subtotal other* | 234 (11) | 70 (10) | 86 (14) | 390 (11) |
| None | | | | |
| Chose to handle it alone‡ | 15 (<1) | 4 (1) | 1 (<1) | 20 (1) |
| No help received§ (n(%)) | 428 (20) | 129 (18) | 141 (23) | 698 (20) |
| Specific support sought following any self-harm postbereavement¶ | | | | |
| None | 70 (63) | 34 (81) | 37 (66) | 141 (67) |
| Friend | 18 (16) | 2 (5) | 8 (14) | 28 (13) |
| Family member | 13 (12) | 3 (7) | 7 (13) | 23 (11) |
| General practitioner | 25 (22) | 5 (12) | 12 (21) | 42 (20) |
| Hospital professionals | 10 (9) | 1 (2) | 5 (9) | 16 (8) |
| Counsellor | 9 (8) | 1 (2) | 4 (7) | 13 (6) |
| Mental health team member | 2 (2) | 0 (0) | 3 (5) | 5 (2) |
| Voluntary sector organisation | 1 (1) | 0 (0) | 0 (0) | 1 (<1) |
| School/college teaching staff | 2 (2) | 0 (0) | 0 (0) | 1 (<1) |

*Categories not mutually exclusive.
†Category included organisations such as the diplomatic service, shipping services (for repatriating the body) and employees at the deceased's bank.
‡Sixteen out of 20 people in this category also endorsed other sources of formal or informal support.
§Category excluded those who had used self-help and those who indicated they had chosen to handle the bereavement alone.
¶In the n=210 individuals who had attempted suicide postbereavement; categories not mutually exclusive.

**Table 3** Estimates of the relationship between support outcomes and bereavement exposure (suicide versus sudden natural death)

| Exposure group | Sudden natural death (n=2106) | | Sudden unnatural death (n=712) | | | | | Suicide (n=614) | | | | |
| --- | --- | --- | --- | --- | --- | --- | --- | --- | --- | --- | --- | --- |
| | Prevalence n (%) | OR (reference) | Prevalence n (%) | Unadjusted OR (95% CI) | p Value* | Adjusted† OR (95% CI) | p Value* | Prevalence n (%) | Unadjusted OR (95% CI) | p Value* | Adjusted† OR (95% CI) | p Value* |
| **Primary outcomes** | | | | | | | | | | | | |
| Receipt of formal support postbereavement | 776 (37) | 1 | 272 (38) | 1.05 (0.88 to 1.27) | 0.548 | 1.28‡ (1.05 to 1.56) | 0.015 | 221 (36) | 0.97 (0.80 to 1.18) | 0.753 | 1.17 (0.94 to 1.44) | 0.155 |
| Receipt of informal support postbereavement | 1396 (66) | 1 | 491 (69) | 1.13 (0.92 to 1.38) | 0.257 | 1.06 (0.86 to 1.33) | 0.553 | 389 (63) | 0.83 (0.68 to 1.02) | 0.083 | 0.79‡ (0.64 to 0.98) | 0.038 |
| **Secondary outcomes** | | | | | | | | | | | | |
| No support postbereavement§ | 428 (20) | 1 | 129 (18) | 0.83 (0.66 to 1.05) | 0.122 | 0.83 (0.65 to 1.07) | 0.149 | 141 (23) | 1.21 (0.97 to 1.52) | 0.097 | 1.21 (0.95 to 1.55) | 0.119 |
| Immediate receipt of support (<1 week) | 913 (43) | 1 | 306 (43) | 0.97 (0.81 to 1.17) | 0.747 | 0.96 (0.79 to 1.17) | 0.660 | 219 (36) | 0.74 (0.60 to 0.90) | 0.002 | 0.73‡ (0.59 to 0.90) | 0.003 |
| Delayed receipt of valuable support (>6 months) | 814 (39) | 1 | 291 (41) | 1.05 (0.88 to 1.27) | 0.575 | 1.10 (0.90 to 1.35) | 0.359 | 271 (44) | 1.26 (1.04 to 1.53) | 0.020 | 1.33‡ (1.08 to 1.64) | 0.008 |
| Use of formal support exclusively¶ | 217/1573 (14) | 1 | 76/558 (14) | 0.98 (0.72 to 1.33) | 0.888 | 1.11 (0.80 to 1.54) | 0.516 | 68/441 (15) | 1.12 (0.82 to 1.55) | 0.454 | 1.26 (0.90 to 1.76) | 0.183 |
| Help sought for postbereavement self-harm†† | 42/112 (38) | 1 | 8/42 (19) | 0.37 (0.15 to 0.95) | 0.038 | 0.43 (0.16 to 1.13) | 0.086 | 19/56 (34) | 0.82 (0.41 to 1.65) | 0.579 | 0.98 (0.44 to 2.17) | 0.953 |

*Significance threshold of p=0.05 for primary outcomes and p=0.01 for secondary outcomes.
†Adjusted for age, gender, socioeconomic status, preloss depression, preloss suicidal and non-suicidal self-harm, other family history of suicide (excluding index bereavement), time since bereavement and kinship to the deceased.
‡Association no longer significant when stigma added to final adjusted model.
§Outcome excluded those who solely endorsed that they chose to handle the bereavement alone.
¶In subset of n=2572 receiving support after bereavement.
††In sub-set of n=210 who had attempted suicide postbereavement.

**Table 4** Estimates of the relationship between support outcomes and bereavement exposure (suicide versus sudden unnatural death)

| Exposure group | Sudden unnatural death (n=712) | Suicide (n=614) | | | | |
|---|---|---|---|---|---|
| | OR (reference) | Unadjusted OR (95% CI) | p Value* | Adjusted† OR (95% CI) | p Value* |
| **Primary outcomes** | | | | | |
| Receipt of formal support postbereavement | 1 | 0.92 (0.72 to 1.16) | 0.462 | 0.91 (0.72 to 1.16) | 0.437 |
| Receipt of informal support postbereavement | 1 | 0.74 (0.58 to 0.95) | 0.020 | 0.74‡ (0.58 to 0.96) | 0.022 |
| **Secondary outcomes** | | | | | |
| No support postbereavement§ | 1 | 1.46 (0.95 to 1.52) | 0.010 | 1.46 (1.09 to 1.95) | 0.011 |
| Immediate receipt of support (<1 week) | 1 | 0.76 (0.60 to 96) | 0.021 | 0.76 (0.60 to 0.97) | 0.025 |
| Delayed receipt of valuable support (>6 months) | 1 | 1.19 (0.94 to 1.51) | 0.139 | 1.21 (0.95 to 1.54) | 0.120 |
| Use of formal support exclusively¶ | 1 | 1.16 (0.79 to 1.70) | 0.463 | 1.13 (0.77 to 1.67) | 0.542 |
| Help sought for postbereavement self-harm†† | 1 | 2.18 (0.79 to 5.98) | 0.131 | 2.28 (0.78 to 6.68) | 0.132 |

*Significance threshold of p=0.05 for primary outcomes and p=0.01 for secondary outcomes.
†Adjusted for age, gender, socioeconomic status, preloss depression, preloss suicidal and non-suicidal self-harm, other family history of suicide (excluding index bereavement), time since bereavement and kinship to the deceased.
‡Association no longer significant when stigma added to final adjusted model
§Outcome excluded those who solely endorsed that they chose to handle the bereavement alone.
¶In subset of n=2572 receiving support after bereavement.
¶In subset of n=210 who had attempted suicide postbereavement.
††In subset of n=210 who had attempted suicide postbereavement.

HEIs and the pronounced male non-response bias limit generalisability beyond highly educated female groups. The limited age-range sampled restricts generalisability beyond young adults. Without a denominator, we were unable to present a response rate but assume that the majority of non-responders were non-bereaved. It was possible that those worst affected had biased recall of support received and its value. Our multivariable models included pre-bereavement depression as a potential confounding variable but did not account for pre-bereavement anxiety or other mental disorders. If those are differentially elevated prior to suicide bereavement, as shown in previous studies,[1] stigma associated with mental illness and/or poor experiences of services might influence receipt of support in this group. Models for two secondary outcomes (exclusive use of formal support; help seeking for attempted suicide postbereavement) lacked sufficient power due to group sizes <466, and larger studies are needed to investigate these hypothesised associations. Despite testing for an interaction with gender, we acknowledge such tests' limited statistical power. Given gender differences in help-seeking for mental illness,[22] particularly in relation to informal support,[23] it would have been desirable to have conducted gender-specific analyses, but this was not possible due to the low numbers of men responding.

## Policy implications

The quarter of our suicide-bereaved sample who received no support represents failed implementation of UK suicide prevention strategies.[4–6] This group was distinct from the 1% who stated that they preferred to cope without assistance. The inequities in informal support we identified for people bereaved by suicide suggest a need for psychosocial interventions to address social avoidance and stigmatising attitudes within social networks. Public education to raise awareness of the vulnerabilities of people bereaved by suicide, the range of support available[36] and advice on how to support them[36 37] could encourage social networks to respond more readily after suicide loss. This, along with interventions to address self-stigma, might also encourage the bereaved to seek help by reinforcing the idea that they are worthy of support. Current UK developments in national systems of early outreach after suicide[38] will address the identified delays in support, particularly at a stage when motivation and awareness is low.[10]

## Further research

Research is needed to explore the influence of stigma on willingness to seek help after suicide bereavement and on others' readiness to offer support. Thematic analysis of our qualitative survey data will permit a more nuanced understanding of this. Studies that deepen our understanding of help-seeking preferences after suicide attempt in people bereaved by suicide might help address risk of reattempt. Expanding the limited evidence base for interventions after suicide bereavement[7] is important,

as is investigating the potential for adverse psychological effects of early[39] and peer support[40] interventions.

## CONCLUSIONS

Our study demonstrated clear inequities in the support received by people bereaved by the suicide of a close contact, manifested in delayed receipt of support and a lesser likelihood of receiving support from family and friends. It is concerning that two-thirds of a group featuring so prominently in UK suicide prevention strategies receive no formal support and that a quarter receive no support at all. Those responsible for implementing suicide prevention strategies should commission lay guidance on how to support someone bereaved by suicide and improve national systems of immediate outreach after suicide loss.

**Acknowledgements** We would like to thank all the HEIs from England, Wales, Northern Ireland and Scotland that consented to participate, listed below, and all the bereaved individuals who took time to respond to the online survey. *Participating HEIs*: Bishop Grosseteste University College Lincoln; Bournemouth University; Central School of Speech and Drama; City University; Cranfield University; Courtauld Institute; De Montfort University; University of Greenwich; King's College London; Liverpool Institute for Performing Arts; Liverpool John Moores University; London Metropolitan University; Norwich University College of the Arts; Royal Veterinary College; School of Oriental and African Studies; St George's, University of London; Staffordshire University; Trinity Laban Conservatoire of Music and Dance; UCL; University Campus Suffolk; University of Bedfordshire; University of Chester; University of Cumbria; University of Leeds; University of Liverpool; University of Oxford; University of Southampton; University of Worcester; University of Westminster; Queen Margaret University; Heriot-Watt University; Scottish Agricultural College; University of Dundee; Cardiff University; Cardiff Metropolitan University (formerly University of Wales Institute Cardiff); Queen's University Belfast and University of Ulster.

**Contributors** ALP, DO and MK conceived and designed the study. ALP, DO, MK, PM and LS contributed to questionnaire design and recruitment strategy. ALP recruited participants, managed the survey and collected data. ALP, KR and LM cleaned data. ALP, KR, LM and DO conducted data analysis. ALP, KR, PM, LS, LM, MK and DO interpreted data, contributed to writing of the report and approved the final version before submission. ALP conducted the literature search. ALP had full access to all the data in the study and takes responsibility for the integrity of the data and the accuracy of the data analysis. No participant is identifiable from the analysis or study report. ALP is the guarantor.

**Ethics approval** UCL Research Ethics Committee.

**Provenance and peer review** Not commissioned; externally peer reviewed.

**Data sharing statement** Informed consent for collecting patient-level data was obtained on the basis that data would be anonymised, stored in accordance with the Data Protection Act 1998, only used for the purposes of the study and not transferred to an organisation outside UCL. Requests to collaborate on analysis of anonymised data with low risk of identification should be made by contacting the corresponding author.

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
