## [Reviewer comments · BMJ Open]

ARTICLE DETAILS

TITLE (PROVISIONAL)	Support received after bereavement by suicide and other sudden deaths: a cross-sectional UK study of 3,432 young bereaved adults
AUTHORS	Pitman, Alexandra; Rantell, Khadija; Moran, Paul; Sireling, Lester; Marston, Louise; King, Michael; Osborn, David

VERSION 1 - REVIEW

REVIEWER	James Bolton University of Manitoba, Canada
REVIEW RETURNED	25-Oct-2016

GENERAL COMMENTS	In this study authors sought to determine the differences in social support received by people bereaved by suicide as well as people bereaved by other causes of sudden death. The study population included 3432 respondents aged 18-40 from a sample of 659,572 individuals at 37 Higher Education Institutions (HEI) in the UK. 61% of the population was bereaved after the age of 10 by natural causes, 21% by sudden unnatural causes and 18% by suicide. Questions and comments: This was a very interesting study that tackles an important question of the support received by people bereaved by suicide. The paper was well-written and the study well designed with some interesting results. That being said there are some things to consider 1) The authors do not provide a rationale for limiting the study sample between 18-40. Given that they sampled both staff and students at a university they are excluding a sample of adults between 40-65 without sufficient rationale. Given that sudden natural deaths are an outcome of interest, and that these would be much more prevalent in older adults (who would have bereaved spouses, friends, and coworkers in the same age group), it is surprising that these were excluded. This should be explained, added to the limitations, and one wonders whether the title should be changed to reflect this.2) It would be interesting to determine whether the age at which bereavement occurred would influence the level of support received. E.g. a bereaved child may be provided different resources compared to a bereaved adult. If services for adolescents and adults are separate/different in those jurisdictions, perhaps a sub-analysis stratifying these age groups?3) Analyses that combined both sudden unnatural deaths and sudden natural death and used this category as the reference group compared to suicide could provide more insight.
--

	4) The consideration of the 8 confounders is admirable, and the choices are good and rooted in previous literature. However, I am surprised that pre-existing depression was the only mental disorder considered. There is literature identifying other mental disorders in the pre-bereavement time period that differentially associate with people who are about to become bereaved versus those that are not. Is there an opportunity to include other disorders? If not, this is a significant limitation as it potentially influences the outcome variables. Should be discussed.
--	--

REVIEWER	Karen Galway Queen's University Belfast
REVIEW RETURNED	20-Dec-2016

GENERAL COMMENTS	Many thanks for the opportunity to review this paper. This is a very timely and important paper illustrating a clear research gap and policy need to implement guidelines addressing the needs of people bereaved by suicide. The introduction provides a strong rationale for the study. The methods are well described, the results are clearly laid out and the discussion contextualises the results well in terms of research, policy and practice. There were three points that might benefit from more clarity or explanation: Page 8 - what does 'excluding index bereavement' mean? Page 9 - presumably running separate statistical models for males and females was not possible due to the imbalance in age of respondents? This should perhaps be stated for clarity as there is such strong literature around help-seeking and gender differences. Page 33 the title is missing from the recruitment flowchart. I do not recall any reference to a flowchart in the text. Please double check that. Many thanks
---

VERSION 1 – AUTHOR RESPONSE

Reviewer 1:

Comment 1) The authors do not provide a rationale for limiting the study sample between 18-40. Given that they sampled both staff and students at a university they are excluding a sample of adults between 40-65 without sufficient rationale. Given that sudden natural deaths are an outcome of interest, and that these would be much more prevalent in older adults (who would have bereaved spouses, friends, and coworkers in the same age group), it is surprising that these were excluded. This should be explained, added to the limitations, and one wonders whether the title should be changed to reflect this.

Response:

We have changed the title to reflect the fact that this was a young sample. Our original aim in conducting this survey had been to focus on young adults because this was the group of greatest

policy interest at the time in UK suicide prevention strategies. We defined the age range of 18-40 based on WHO definitions and publications (using our systematic review, ref 22). Another reason for restricting the age range was that it reduced the recall period to a maximum of 30 years, minimising the potential for memory decay, as well as narrowing the period effect from cultural change. We have explained this more fully in our Methods section to justify the rationale for limiting the study sample to age 18-40, and amended our Discussion to highlight the limited generalisability of findings beyond this age group. We have also edited the strengths and limitations bullet points to demarcate limits of generalisability.

We have amended the following sentences in our Introduction: (p 4-5)

“We focused on young adults given concerns about their vulnerabilities to suicide(22), their tendency to avoid accessing mental health services(23), and their priority status within UK suicide prevention strategies(4-6). Surveying this age range also minimised the potential for memory decay, and narrowed period effects”.

We have included the following sentence in our Discussion: (p 13)

“The limited age-range sampled restricts generalisability beyond young adults”.

Comment 2) It would be interesting to determine whether the age at which bereavement occurred would influence the level of support received. E.g. a bereaved child may be provided different resources compared to a bereaved adult. If services for adolescents and adults are separate/different in those jurisdictions, perhaps a sub-analysis stratifying these age groups?

Response:

Across the UK different geographical areas use different cut-offs for child and adolescent services, so as a group we agreed that the most consistent transition would be at age 18. We added an interaction term to test whether being a child or adult at the time of the bereavement influenced receipt of support, but this did not modify associations.

We have amended the following sentence in our Results: (p 11)

“Interaction tests showed no evidence that gender, or childhood versus adult bereavement, modified any of the associations identified”.

Comment 3) Analyses that that combined both sudden unnatural deaths and sudden natural death and used this category as the reference group compared to suicide could provide more insight.

Response:

We have added this as a post hoc sensitivity analysis, and found very similar associations in terms of magnitude and direction, apart from one of the secondary outcomes (delayed receipt of valuable support), which was no longer significant. As in the main analysis, all significant associations became non-significant when accounting for high stigma scores.

We have included the following sentence in our Results: (p 11)

“In an analysis comparing suicide bereavement to all non-suicide sudden bereavements we found

similar associations, in terms of magnitude and direction, apart from the association of suicide bereavement with one secondary outcome (delayed receipt of valuable support), which became non-significant”.

Comment 4) The consideration of the 8 confounders is admirable, and the choices are good and rooted in previous literature. However, I am surprised that pre-existing depression was the only mental disorder considered. There is literature identifying other mental disorders in the pre-bereavement time period that differentially associate with people who are about to become bereaved versus those that are not. Is there an opportunity to include other disorders? If not, this is a significant limitation as it potentially influences the outcome variables. Should be discussed.

Response:

We agree that it would have been helpful to have explored the confounding effects of other pre-bereavement psychiatric disorders, but we lack these variables because we used data from an internet-based survey. The purpose of this sampling method was to access hard-to-reach groups, and enhance disclosure of suicidal behaviour, particularly that unreported to health professionals. However this mode of data collection created a risk of response fatigue and attrition, particularly with a long questionnaire. We therefore had to restrict the number of measures included, and were unable to include screens for the full range of individual mental disorders. We prioritised depression as a key confounder and used a validated measure (the 2-item CIDI screen), clarifying whether this was before or after the bereavement. In light of the findings of Bolton et al 2013, and other studies summarised in our 2014 systematic review (ref 1), it would have been useful to have included a validated measure of pre-bereavement anxiety. We have added to the limitations that our findings may be confounded by unmeasured mental health problems prior to the bereavement.

We have included the following sentence in our Discussion: (p 13-14)

“Our multivariable models included pre-bereavement depression as a potential confounding variable but did not account for pre-bereavement anxiety or other mental disorders. If those are differentially elevated prior to suicide bereavement, as shown in previous studies(1), stigma associated with mental illness and/or poor experiences of services might influence receipt of support in this group”.

Reviewer 2:

Many thanks for the opportunity to review this paper. This is a very timely and important paper illustrating a clear research gap and policy need to implement guidelines addressing the needs of people bereaved by suicide. The introduction provides a strong rationale for the study. The methods are well described, the results are clearly laid out and the discussion contextualises the results well in terms of research, policy and practice.

There were three points that might benefit from more clarity or explanation:

Page 8 - what does 'excluding index bereavement' mean?

Response:

This referred to any family history of suicide contributed by an index suicide bereavement of a family member. We have clarified this in the text by amending this to index suicide bereavement (p 8).

Page 9 - presumably running separate statistical models for males and females was not possible due

to the imbalance in age of respondents? This should perhaps be stated for clarity as there is such strong literature around help-seeking and gender differences.

Response:

We conducted an interaction test for gender and found that it did not modify our associations. We have now added to our Discussion an acknowledgement of the limited power of interaction tests, and our inability to conduct separate analyses for men and women due to the relatively small numbers of male respondents.

We have included the following sentences in our Discussion: (p 14)

“Despite testing for an interaction with gender, we acknowledge such tests’ limited statistical power. Given gender differences in help-seeking for mental illness(22), particularly in relation to informal support(23), it would have been desirable to have conducted gender-specific analyses but this was not possible due to the low numbers of men responding.”

Page 33 the title is missing from the recruitment flowchart. I do not recall any reference to a flowchart in the text. Please double check that.

Response:

The legend was missing – thank you for pointing that out. It has now been added within the list of supplementary files.

VERSION 2 – REVIEW

REVIEWER	Karen Galway School of Nursing and Midwifery, Queen's University Belfast, United Kingdom
REVIEW RETURNED	27-Feb-2017

GENERAL COMMENTS	This is a revised version. All points of concern have been fully addressed by the authors. The paper is in excellent shape for publication.
---